# Robot-assisted, source-camera-coupled multi-view broadband imagers for ubiquitous sensing platform

Kou Li [1], Ryoichi Yuasa[1], Ryogo Utaki[1], Meiling Sun[1], Yu Tokumoto[1], Daichi Suzuki[2] & Yukio Kawano [1 ✉]

Multi-functional photo-imaging garners attention towards the development of universal safety-net sensor networks. Although there are urgent needs to comprehensively address the optical information from arbitrarily structured and located targets, investigations on multi-view sensitive broadband monitoring, being independent of the operating environment, are yet to be completed. This study presents a robot-assisted, photo-source and imager implanted, multi-view stereoscopic sensitive broadband photo-monitoring platform with reflective and transmissive switchable modes. A multifaceted photo-thermoelectric device design based on flexible carbon nanotube films facilitates the prototype demonstrations of non-destructive, target-structure-independent, free-form multi-view examinations on actual three-dimensional industrial components. Further functionalisation, namely, a portable system utilising three-dimensional printing and ultraviolet processing, achieves the unification of freely attachable photo-imagers and miniature photo-sources, enabling location-independent operation. Consequently, the non-destructive unmanned, remote, high-speed, omni-directional testing of a defective aerial miniature model winding road-bridge with a robot-assisted photo-source imager built into a multi-axis movable photo-thermoelectric monitor arm is demonstrated.

[1] Laboratory for Future Interdisciplinary Research of Science and Technology, Tokyo Institute of Technology, Tokyo, Japan. [2] Center for Emergent Matter Science, RIKEN, Saitama, Japan. ✉email: kawano@pe.titech.ac.jp

Non-destructive sensing and imaging techniques based on electromagnetic (EM) waves have garnered significant attention in both industry and academia. This is because of increasing demands for better performance, safety and quality assurance of industrial components, especially in the impending Internet of Things (IoT) society[1,2]. In particular, this situation requires multifaceted perspectives and comprehensive investigations of sensing devices dealing with composite multi-layered target objects of arbitrary structures or locations. Previous studies have reported rapid progress in imaging modalities by widening their application in certain regions of the EM spectrum, including visible light[3,4], infrared (IR)[5,6], terahertz (THz) and sub-THz wave[7-9] and millimetre-wave (MMW)[10,11]. Broadband photo-monitoring has been proven to be effective specifically in aggregating multi-spectral optical properties and information to test the composite material and layer structure[12,13]. Concurrently, bendable, flexible and easily attachable imager devices facilitate multi-functional sensing. Specifically, flexible and wearable scanner sheets have reflective and transmissive switchable modes that are comfortably applicable to any three-dimensional (3D) curvature through an adjustment of the device configuration with target structures. Moreover, the free-form imager setup yields multi-view stereoscopic visualisation[14], which can also be successfully applied in both flat and bent samples without forming blind spots. Additionally, the portability of sensing modules must be addressed and considered to overcome the limitation posed by the operation locations. The built-in implementation of photo-imagers and photo-sources is essential for designing a compact sensing module[15]. Here, uncooled device operation is indispensable for mobile systemisation and use in practical social scenes, being free from bulky cooling equipment. Moreover, the robot-assisted module can potentially govern ubiquitous sensing, where the unmanned remote high-speed photo-monitoring, which is independent of the operation environment, is feasible[16]. The transition from manned to robotic inspection can make operation safer and more sustainable. Some examples of robotic operation include disconnection testing of power-transmission lines with aerial modules, crack examination of sea bridges with wall-climbing units and exploring cramped environments with self-driving systems.

Thus, further efforts are needed to enable photo-imagers, which play an essential role in ubiquitous sensing platforms, with the following characteristics:

(i) Flexible switching of the multi-view stereoscopic system with reflective and transmissive sensing options with a proper choice of a freely attachable uncooled broadband photo-absorbent thin film.

(ii) Built-in implementation of miniature photo-sources in flexible multi-view stereoscopic photo-detector frameworks.

(iii) Target-location-independent, high-speed and omnidirectional photo-monitoring via robot-assisted module driving.

To this end, several studies have investigated diverse systems, photo-sources and photo-imagers including functional unmanned remote sensing robotics[17,18], high-usability miniature photo-sources[19,20], flexible multi-view stereoscopic photo-imagers[21,22] and highly efficient uncooled broadband photo-absorbent materials[23,24]. In a related study, Yang et al. developed the thermo-phototronic effect[25], which is based on the combination of thermoelectric effect with the photoelectric effect in some semiconductor materials[26-29]. However, their functional integrations have not been adequately verified, hindering the fruition of a ubiquitous sensing platform.

In this study, we develop a robot-assisted, photo-source and imager implanted, multi-view and sensitive broadband photo-monitoring platform with switchable reflective and transmissive modes, demonstrating the aforementioned characteristic requirements. The proposed module employs (I) the freely attachable photo-thermoelectric (PTE) technique on uncooled sensitive broadband photo-imager sheets with physically and chemically enriched carbon nanotube (CNT) thin flexible films. The freely attachable photo-imager sheet enables the switching from reflective to transmissive modes and offers multi-view stereoscopic sensing operation. This free-form photo-monitoring not only allows comprehensive and non-destructive inspections of actual curvilinear industrial components (beverage bottles, gas or water pipes) but also provides a unique opportunity for the remote arbitrary hierarchical image extraction of multi-layered 3D structures via multi-frequency band sensing (sub-THz, near-infrared (NIR)). Simultaneously, the effective utilisation of the 3D printer and ultraviolet (UV) laser facilitates (II) the embedding of high-output-power miniature photo-sources. The resulting photo-source-implanted compact sensing module performs a portable 360°-view photo-monitoring. Finally, the portability of the photo-monitoring system leads to the built-in implementation of the present sensing module in a multi-axis movable-arm unit. Consequently, (III) the robot-assisted module operation verifies the non-destructive, unmanned, remote, high-speed and omni-directional photo-monitoring of the miniature model of an aerial defective winding road-bridge. These efforts potentially provide a roadmap for the materialisation of a ubiquitous sensing platform. Our results and the concept of this study could also be sued for a sustainable, long-term operable and user-friendly IoT system of a sensor network, including permanent and regular in situ inspections at construction sites and in line monitoring of fine-processed industrial products.

## Results

**Fundamental device design for a CNT-film flexible broadband PTE photo-imager**. We discuss the role of fundamental property of the uncooled flexible broadband photo-imager in the development of the proposed sensing module. Among the various potential photo-detection mechanisms, the PTE technique is chosen because it facilitates the proposed scheme via advantageous broadband operation compared to electronic-type detectors, and the advantageous uncooled characteristic of the photo-imager compared to photon-type devices[30]. Several studies have reported on different materials for functional photo-imagers, including graphene[31], CNT[32], $MoS_2$[33], SnSe[34], $NbS_3$[35] and PEDOT:PSS[36]. Among them, randomly stacked single-walled CNT (SWCNT) films are employed as the device channels in this study; this is because, when compared to the aforementioned materials, the CNT films collectively demonstrate high mechanical strength[37], flexibility[38] and advantageous excellent uncooled broadband photo-absorption[39]. Moreover, the Seebeck coefficients of CNT films typically range from tens to hundreds of μV/K[40-42]. Given these multi-functional properties, CNT films allow for efficient broadband operation (while showing good PTE properties), efficient conversion of photo-irradiation power to electrical signals, and flexible adjustment and conformance to target structures. However, the photo-detection sensitivity of the existing PN-junction type CNT-film device (noise equivalent power (NEP) of 30 $pWHz^{-1/2}$)[43] is still lower than that of state-of-the-art, solid-state, uncooled, broadband photo-detectors (sub-10 $pWHz^{-1/2}$)[44,45]. This demonstrates a need to understand the device's working mechanism.

We first tackled the multifaceted device design of CNT-film flexible broadband photo-imagers. Figure 1a and Supplementary Fig. 1 illustrate the working principle of the liquid-coated N-type chemical-carrier doping on originally P-type SWCNT films, and the resulting photo-detection based on the PTE effect. Here, a semiconducting and metallic mixed-type SWCNT film was

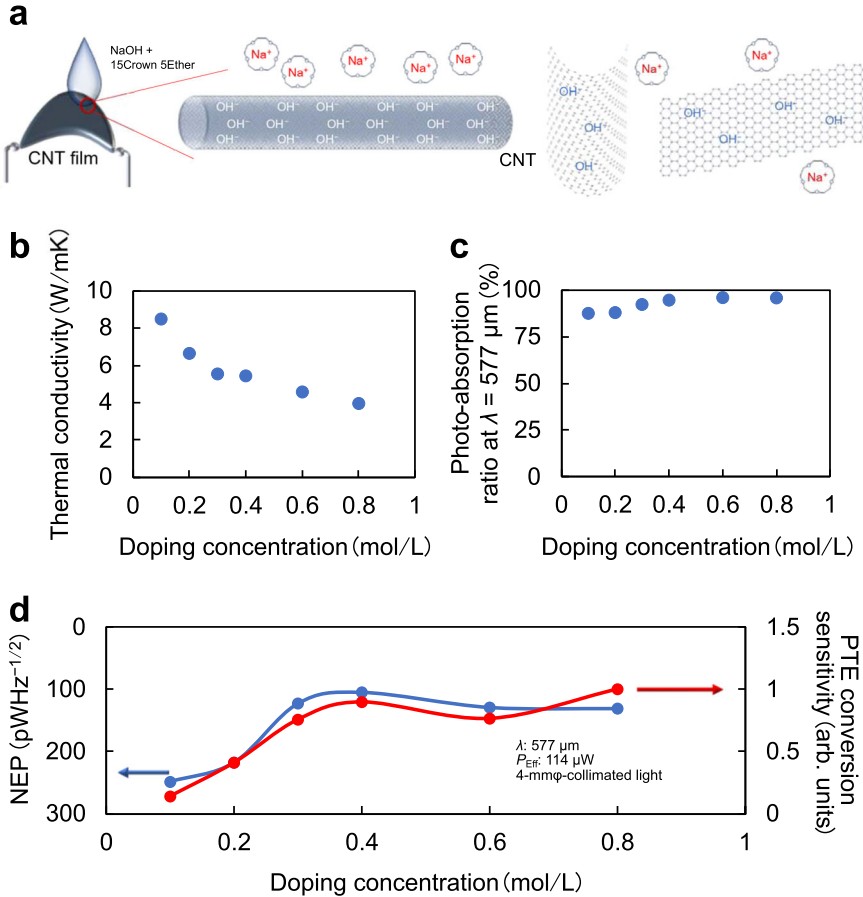

**Fig. 1 Photo-thermoelectric (PTE) conversion tuning in response to concentration of N-type chemical-carrier dopant. a** Schematic of N-type doping on P-type carbon nanotube (CNT) films. The change in dopant concentration resulted in the tuning of each PTE parameter including **b** thermal conductivity, **c** photo-absorption ratio ($\lambda = 577 \, \mu m$), Seebeck coefficient (Supplementary Fig. 4a), and electrical resistance (Supplementary Fig. 4b). **d** Change in the noise equivalent power (NEP) value and the PTE conversion sensitivity vs. the N-type doping concentration. The experimentally obtained minimum NEP values of the proposed CNT-film photo-imager at broad wavelength bands (millimetre-wave (MMW)—near-infrared (NIR)) are as follows: 236 pWHz$^{-1/2}$ ($\lambda = 1.15$ mm, $V = 3.26$ mV, $P_{Eff} = 256 \, \mu W$, Beam spot: 10-mmφ-collimated), 105 pWHz$^{-1/2}$ ($\lambda = 577 \, \mu m$, $V = 328 \, \mu V$, $P_{Eff} = 114 \, \mu W$, Beam spot: 4-mmφ-collimated), 30 pWHz$^{-1/2}$ ($\lambda = 300 \, \mu m$, $V = 598 \, \mu V$, $P_{Eff} = 5.97 \, \mu W$, Beam spot: 4-mmφ-collimated), 8.57 pWHz$^{-1/2}$ ($\lambda = 10.3 \, \mu m$, $V = 13.8$ mV, $P_{Eff} = 39.3 \, \mu W$, Beam spot: 1 mmφ), and 11.1 pWHz$^{-1/2}$ ($\lambda = 870$ nm, $V = 421 \, \mu V$, $P_{Eff} = 1.55 \, \mu W$, Beam spot: 5-mmφ-collimated), for the device with an electrical resistance of 555 Ω.

employed as the photo-imager channel material because of its superior PTE properties compared to those of multi-walled or semiconducting-separated single-walled CNT films (Supplementary Fig. 2). The photo-induced PTE response at the PN interface can be described by the following equation:[43]

$$\triangle V = \left( S_{P-CNT\,film} - S_{N-CNT\,film} \right) \times \triangle T \qquad (1)$$

where $\triangle V$ is the direct current (DC) voltage photo-response, $S_{P-CNT\,film}$ is the Seebeck coefficient of the P-type CNT film, $S_{N-CNT\,film}$ is the Seebeck coefficient of the N-type CNT film, and $\triangle T$ is the photo-induced thermal gradient across the CNT-film channel. In this study, the PN junction of the CNT film has been employed in the proposed device structure to facilitate higher intensity photo-response detection compared with that of the channel-electrode interfaces (Supplementary Fig. 3). Each Seebeck coefficient was measured as $S_{P-CNT\,film}$: 47 μV/K, $S_{N-CNT\,film}$: −42 μV/K, $S_{Electrode}$: 1.5 μV/K, and the effective Seebeck coefficient of the PN junction can be maximised ($S_{P-CNT\,film} - S_{N-CNT\,film}$) while those of each electrode junction are suppressed ($|S_{Electrode} - S_{P-CNT\,film}|$, $|S_{N-CNT\,film} - S_{Electrode}|$). Equation 1 shows that optimising the chemical-carrier doping on the CNT films, where respective $S_{CNT\,film}$ and $\triangle T$ (derived from

thermal conductivity) can be affected[46], plays an important role in enhancing the photo-detection sensitivity. To overcome the insufficient photo-detection sensitivity of PN-junction type CNT films compared with those of the aforementioned solid-state devices, each material property of chemically N-doped CNT films was assessed. Figure 1b–c, and Supplementary Fig. 4a–d map the changes in the PTE parameters: (1) the Seebeck coefficient, (2) thermal conductivity, (3) electrical resistance and (4) photo-absorption ratio (at $\lambda = 577/300 \, \mu m$) of the N-type-doped CNT films in response to change in the dopant concentration. These changes allowed us to tune the PTE conversion sensitivity (see PTE conversion sensitivity evaluation in the Methods section). Simultaneously, we observed a change in NEP values with varying dopant concentration, and both showed similar tendencies (Fig. 1d and Supplementary Fig. 4e). The presented tuning of the NEP values and PTE conversion enabled us to select doping concentration regions with higher photo-detection sensitivity (we set 0.4 mol/L in the following investigations).

Additionally, as seen from Eq. 1, a low-loss thermal gradient $\triangle T$ design enables better photo-detection performance. Indeed, the choice of adequate thermal conductivity for the substrate prevented the thermal degradations of both the photo-induced

heating in the CNT-film channels and the incidental photo-responses. Supplementary Fig. 5a–c shows the increase in magnitude of the photo-responses (up to 30 times) among substrate candidates, which are the flexible cellulose acetate membrane and rigid silicon. Together with the previous channel-size-based $\triangle T$ maximisation[47], the physical and chemical enrichments of CNT-film flexible broadband photo-imagers resulted in a minimum uncooled non-vacuum NEP value of 8.57 pWHz$^{-1/2}$ ($\lambda = 10.3\,\mu m$), which is comparable with that of the aforementioned solid-state devices. Concurrently, the evaluation of the unique device properties, including the beam-shape dependence on the photo-responses (Supplementary Fig. 6a–e), stability of the PTE conversion against the whole device folding (Supplementary Fig. 7a–b), wideband photo-detection (sub-THz ~ NIR, Supplementary Fig. 8a), noise suppression (Supplementary Fig. 8b), simple photo-imaging demonstration (Supplementary Fig. 8c) and time constant (Supplementary Fig. 9), highlights

the potential of the proposed photo-imager for applications in multi-view stereoscopic broadband sensing modules.

**A reflective multi-view stereoscopic capsule PTE imager.** Based on this PTE device design for the CNT-film flexible broadband photo-imager, we developed the multi-view stereoscopic sensing modules with reflective and transmissive switchable modes. Figure 2a depicts a reflective multi-view stereoscopic capsule PTE imager. The proposed module consists of a 3D-printed hemicylindrical substrate in which a freely attachable CNT-film PTE image sensor array sheet (Supplementary Figs. 10, 11a–b, 12a–b) was patched. The waveguide fibre from external photo-sources was embedded in the window frame of the substrate. Reflective multi-view stereoscopic capsule photo-imaging can be performed while scanning target samples in an arc, as shown in Supplementary Fig. 13a. Supplementary Fig. 13b demonstrates the

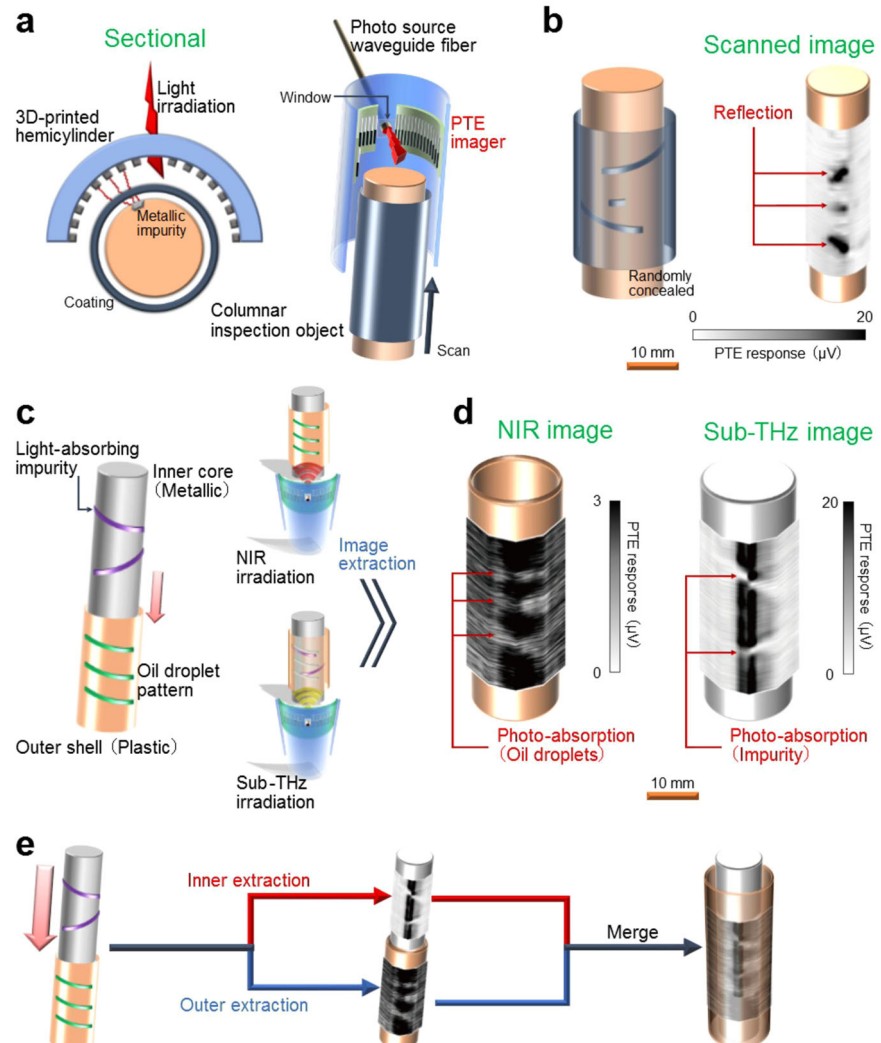

**Fig. 2 Reflective multi-view stereoscopic capsule photo-thermoelectric (PTE) imager. a** Schematic of the module and its fundamental operation mechanism. Operation of the reflective multi-view stereoscopic capsule PTE imager of the Methods section describes the corresponding detailed conditions. **b** Non-destructive reflective multi-view stereoscopic capsule photo-imaging inspection of a glass beverage bottle (in the sub-terahertz (sub-THz: $\lambda = 1.15\,mm$) frequency region) in which metallic impurities were concealed by an opaque coating. The impurities were visualised via detecting local reflection signals corresponding to their locations. **c** Remote hierarchical image extraction of a multi-layered columnar object via the reflective multi-view stereoscopic capsule multi-frequency band sensing. The figure represents the measurement setup in sub-THz ($\lambda = 1.15\,mm$) and near-infrared (NIR: $\lambda = 870\,nm$) frequency regions. The module driving in these two bands performed image extractions of **d**, the outer shell (NIR), and the inner core (sub-THz). We employed 30-μm-thick SWCNT films as the light-absorbing impurities. **e** Simple image restoration of the multi-layered columnar object by covering the inner hierarchical image with the outer hierarchical one. Here, the multi-view stereoscopic PTE images were scanned at a speed of 10 Hz (**b, d**).

smooth handling of columnar objects by the proposed module under an effective viewing angle of ~60°. Figure 2b, and Supplementary Figs. 13c–e and 14a–b show the non-destructive, reflective, multi-view stereoscopic capsule photo-monitoring of defective non-metallic and metallic beverage bottles. Various miniscule concealed defects, such as metallic impurities, breakages and surface scratches, were clearly visualised by detecting the local change of reflection signals at each defect. Simultaneously, the results revealed that the proposed module is functional in the broadband frequency region, ranging between the THz and the IR bands. This characteristic further led to the remote hierarchical image extraction of a multi-layered columnar object, as shown in Fig. 2c–d. Here, a multi-layered column consisting of a metallic inner core and a thin plastic outer shell was employed as a target. The multi-frequency band photo-monitoring yielded the non-destructive image extraction of the outer surface via the NIR region and the inner structure via the sub-THz region. The success of this technique can be attributed to the optical properties of each target layer (inner: light-absorbing impurities on a metallic column, outer: oil droplets on a plastic shell) in each respective IR and THz band. Specifically, the transmittance difference through the oil droplets shown in Supplementary Fig. 14c (NIR irradiation ($\lambda = 870$ nm): 44% transparency, sub-THz irradiation ($\lambda = 1.15$ mm): 95% transparency) is thought to be effective. Therefore, the sub-THz irradiation could be utilised to monitor the inner structure without interference at the outer surface even in the presence of adhered obstacles such as oil droplets. In contrast, the less permeable NIR image captures both the faint reflection signals on the surface of the plastic shell and their local attenuation due to the photo-absorption at the oil droplets. Thus, the inner and outer image extractions facilitate the simple image reconstruction of the multi-layered columnar object (Fig. 2e).

These suggest that the proposed module plays a particularly important role in the ubiquitous sensing of industrial columnar components (power-transmission lines, communication cables), which require high quality and safety standards. Our multi-frequency band photo-monitoring technique also provides an opportunity for the arbitrary hierarchical image extraction of multi-layered 3D objects via the appropriate selection of sensing frequencies.

**A transmissive multi-view stereoscopic PTE endoscope.** Following the reflective photo-monitoring discussed earlier, we designed a transmissive multi-view stereoscopic PTE endoscope (Fig. 3a) as another type of switchable reflective and transmissive modes for the proposed module. The proposed module consists of a 3D-printed columnar supporter, which was wrapped in the freely attachable photo-imager sheet. Scanning cylinder objects while irradiating the proposed module with the external photo-source allowed us to perform the transmissive multi-view stereoscopic endoscopy. In particular, Supplementary Fig. 15a presents the seamless handling of circular cylindrical objects by the proposed module, under an effective viewing angle of ~80°. Figure 3b–c and Supplementary Fig. 15b–d demonstrate the non-destructive transmissive multi-view stereoscopic endoscopy of defective plastic water pipes and an actual gas pipe, respectively. Various concealed minor defects such as a light-absorbing impurity, breakages and surface scratch were clearly visualised by detecting the local change of transmission signals on each defect. The presented endoscopic technique shows the contribution of the proposed module toward the mitigation of the aftermath of natural disasters (i.e. power outs (gas pipes) and water cuts (water pipes) due to the earthquake). Additionally, this endoscopy concept collectively satisfies the multi-view broadband

photo-monitoring. This could bolster existing investigations of endoscopic imagers[48–50] in which these functional consolidations have not been fully tackled.

Concurrently, the proposed robot-assisted free-form photo-monitoring modules can provide the sensing platform with manoeuvrability and easy access compared to manned inspections. They can be applied to winding tunnels, aerial road-bridges and overhead power-transmission lines. To this end, we verified the non-destructive unmanned photo-monitoring of a confined defective miniature L-shaped tunnel via simple robot-assisted self-driving endoscopy (Fig. 3d–e). A commercial self-driving unit covered in the freely attachable photo-imager sheet, smoothly explored the interiors of the tunnel with discretely spread breakages, while the entire outer tunnel-surface was irradiated with the FIR light. In line with the ongoing progress in robotics investigations involving the use of multi-axis movement[51], wall climbing[52], underwater or aerial swimming[53,54] and other operations, this concept of PTE robotics significantly facilitates the diversification of the multi-view broadband photo-monitoring modules.

**Built-in implementation of miniature photo-sources in the sensing module for portable robot-assisted omni-directional operation.** The realisation of free-form multi-view stereoscopic PTE monitoring robotics requires the unification of photo-sources and sensing modules. Specifically, the use of large external photo-sources makes the system immobile, thereby limiting the application of the module mentioned. To address this limitation, we performed the built-in implementation of multiple miniature photo-sources in the free-form sensing module: photo-source-implanted portable 360°-view stereoscopic PTE imager (Fig. 4a). Figure 4b–c demonstrates the built-in implementation of six miniature NIR light emitting diodes (NIR LEDs) in the cylindrical photo-imager and the portable reflective 360°-view stereoscopic surface visualisation of the inserted columnar objects. Metallic patterns on the outer surface of a plastic column, such as discrete impurities and spiral wiring (assuming simple aerial bare power-transmission lines on high-voltage electricity pylons), were omni-directionally captured in the obtained 360° images. Currently, the proposed module can acquire 360° PTE images at a speed of 10 Hz in a simple and portable manner. In particular, rotational scanning is not required since the around-view on targets can be covered with multiple CNT-film pixels. Hence, the measurement time was 24 times shorter than that of single-pixel scanning with the same spatial resolution (600 s → 25 s with or without rotation, with a scan interval of 100 µm). The effective utilisation of the 3D printer and UV laser was indispensable to the flexible design on the window frames (processing resolution: 50 µm for the 3D printer, 10 µm for the UV laser). The window frames were designed to match the size of 2 mmφ × 3 mm height of the NIR LEDs, to tightly embed those miniature photo-sources. This potentially facilitates the built-in implementation of other types of miniature photo-sources, together with recent progress in portable photo-sources, such as quantum cascade laser (Mid-IR)[55], resonant tunnelling diode (THz)[56] and Gunn diode chip (sub-THz, MMW)[57]. In particular, in situ non-destructive omni-directional photo-monitoring of aerial distribution lines (the inner spiral metallic wire bundle was coated with a rubber insulator with a thickness of a few mm) can be performed[58] by employing the IMPATT and Gunn diodes.

Finally, by incorporating the above findings and technologies, we performed a non-destructive unmanned remote high-speed omni-directional inspection of a miniature aerial defective winding road-bridge, with a robot-assisted photo-source-imager built-in multi-axis movable PTE monitor arm. This module

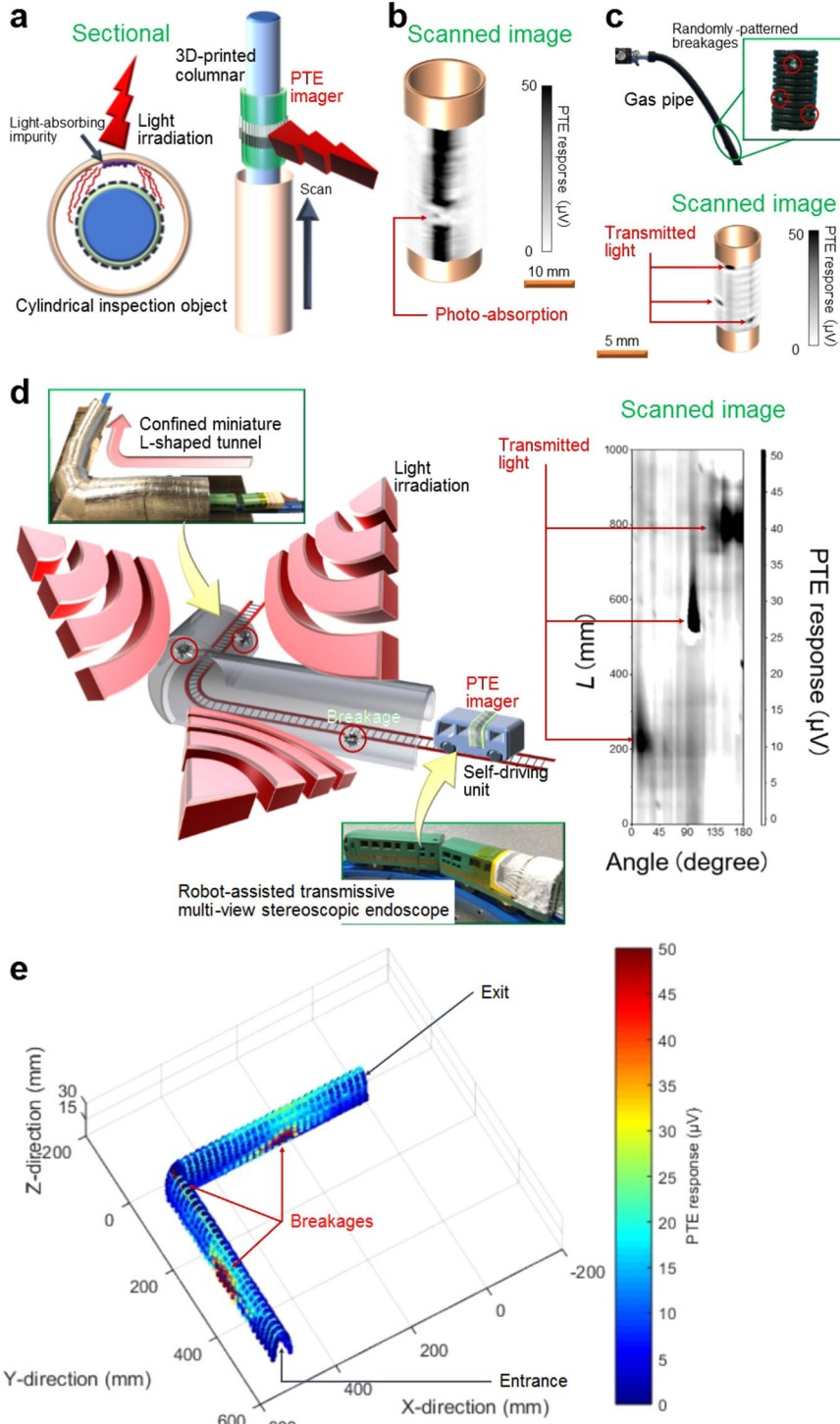

**Fig. 3 Transmissive multi-view stereoscopic photo-thermoelectric (PTE) endoscope. a** Schematic of the module and its fundamental operation mechanism. Operation of the transmissive multi-view stereoscopic PTE endoscope of the Methods section describes the corresponding detailed conditions. Supplementary Fig. 16 shows an evaluation of the fundamental endoscopy performances in response to the target size, shape and structure. **b** Non-destructive endoscopy of a plastic water pipe (in the sub-terahertz (sub-THz: $\lambda = 1.15$ mm) frequency region) in which the light-absorbing impurity was concealed. The impurity was visualised by detecting the locally attenuated transmission signals due to the photo-absorption, which corresponds to its location. **c** Non-destructive endoscopy of an actual gas pipe (in the far-infrared (FIR: $\lambda = 10.3$ μm) frequency region) with discretely spread breakages. The breakages were visualised by detecting local transmission signals corresponding to their locations. The multi-view stereoscopic PTE images were scanned at a speed of 10 Hz (**b**, **c**). **d** Self-driving transmissive multi-view stereoscopic PTE endoscope. Non-destructive unmanned remote endoscopy of a defective miniature L-shaped tunnel with discretely spread breakages was performed. We utilised multiple broadband FIR frequency radiators to illuminate the entire outer surface of the L-shaped tunnel. The breakages were visualised by detecting local transmission signals corresponding to the locations of the breakages. The self-driving endoscope was operated at a speed of 10 mm/s. *L*: scanning direction. **e** Three-dimensional PTE image reconstruction of the target tunnel. The experimentally obtained data were processed via the MATLAB software.

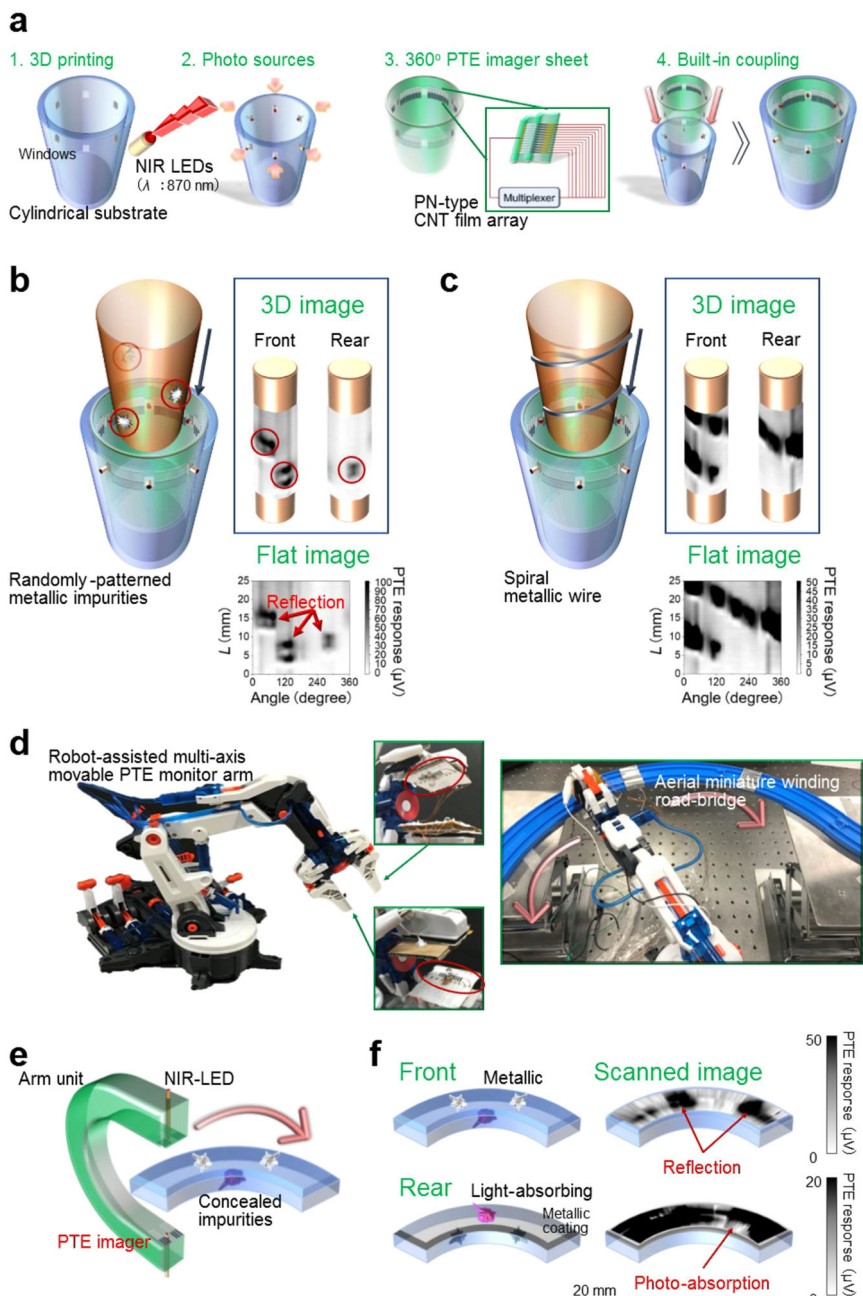

**Fig. 4 Photo-source-implanted portable 360°-view three-dimensional (3D) photo-thermoelectric (PTE) imager for the robot-assisted monitoring applications. a** Fabrication process of the module. Supplementary Fig. 17 and Operation of the photo-source-implanted portable 360°-view stereoscopic PTE imager of the Methods section describe the corresponding detailed operation mechanism and conditions. **b** Portable around-view photo-monitoring of a plastic column (in the near-infrared (NIR: $\lambda = 870$ nm) frequency region) with discretely spread metallic impurities. The impurities were visualised by omni-directionally detecting the local reflection signals corresponding to their locations. L: scanning direction. **c** Portable around-view photo-monitoring of a plastic column (in the NIR frequency region) wrapped by a spiral metallic wire assuming power-transmission line inspections. The metallic wire surface pattern was visualised by omni-directionally detecting the local reflection signals corresponding to its location. Here, the 360°-view stereoscopic PTE images were scanned at a speed of 10 Hz (**b**, **c**). **d** Robot-assisted photo-source-imager built-in multi-axis movable omni-directional PTE monitor arm. We employed a commercial hydraulic 5-axis movable-arm unit. The miniature model of an aerial defective winding road-bridge was set at a height of 50 cm. **e** Fundamental module operation mechanism. A 30-μm-thick carbon nanotube (CNT) film was utilised as the light-absorbing impurity on the road-bridge. We implanted the miniature NIR light emitting diode (NIR LED: $\lambda = 870$ nm, as in **a**) and an 8-pixel freely attachable CNT-film PTE photo-imager array sheet on both upper and lower arm grips. **f** Non-destructive unmanned remote high-speed omni-directional photo-imaging inspection of the aerial road-bridge. Defects on the aerial road-bridge: metallic impurities on the plastic front surface and the light-absorbing impurity on the metallic rear surface were visualised by collectively detecting both local reflection signals corresponding to the locations of the metallic impurities, and the locally attenuated reflection signals due to the photo-absorption corresponding to the location of the light-absorbing impurity. The multi-axis PTE monitor arm was operated at a speed of 15 mm/s.

consists of a hydraulic 5-axis movable-arm unit, where both the miniature NIR LEDs and the freely attachable photo-imager sheets were implanted in its grips (Fig. 4d). Figure 4e–f illustrates the photo-monitoring demonstration. Defects (front: metallic impurities, rear: light-absorbing impurity) were visualised by collectively detecting the local change of reflection signals on each defect, for both the front and real surfaces. Here, the mobility of the proposed module, namely up, down, left, right, forth and back motion, swing and rotary motion and grasping action, allowed the monitor grips to be utilised for the tight sandwiching of the aerial road-bridge. The flexible and smooth control of their spatial positions along the contour and location of the target was also enabled. The presented concept leverages the advantages of free-form photo-monitoring with enriched CNT-film photo-imagers, built-in implementation of miniature photo-sources, and robot-assisted module driving, respectively. These combine to produce unique high-manoeuvrability multi-view stereoscopic sensitive broadband PTE robotics. Additionally, we anticipate a deeper study of dust[59], moisture[60] and corrosion[61]-proof conformal coatings, which further contributes to the practical uses of a ubiquitous sensing platform (e.g. long-term outdoor works regardless of the season and weather).

## Discussion

The findings in this study highlight the necessity of tackling the several unexplored topics on this area, enhancing the presented underlying technologies to practically systemise the proposed ubiquitous sensing platform. One noteworthy issue is that of quantifying the trade-off between photo-monitoring scan speed and acquired image quality, which has not been yet clarified on the proposed experimental setup. We expect that by verifying this factor, we will be able to establish a criterion to select the types of robot-assisted module driving, depending on the target object structures and location. Larger-area pixel configuration (e.g. 2D-matrix structure) is thought to accelerate the system operation speed. The current system operation speed of 10 Hz is bottlenecked by combined use with digital stepping motors, so that 360°-view PTE images can be acquired at 200 Hz, which corresponds to the fundamental device response time (Supplementary Fig. 9), by motionless large-area imaging. Concurrently, a deeper understanding of the optical properties of target materials, such as concrete blocks and polymer resins, is essential for developing a broadband spectral database that potentially facilitates well-substantiated and highly reliable sensing system operation. Unlike the surfaces of defective samples, which mainly require visible light inspections[62,63], the non-destructive photo-monitoring methods of inner defects (i.e. internal breakages, chloride and liquid impurities and corrosion) are still controversial. Hence, systematisation of the optical spectral information on the inner structure, ranging between the MMW, sub-THz and THz frequency bands, can assure greater inspection accuracy under in situ sensing module operation.

In conclusion, this study proposed a robot-assisted, photo-source and imager implanted, and multi-view stereoscopic sensitive broadband photo-monitoring platform with switchable reflective and transmissive modes to achieve a ubiquitous safe and high-quality sensor network. While industrial non-destructive photo-monitoring techniques, being independent of the operating environment, have been in constant demand, we provide a consistent approach from fundamental property assessments of device materials to verifications of the unmanned remote sensing. Specifically, free-form photo-monitoring with enriched CNT-film flexible broadband PTE photo-imager sheets (minimum NEP of 8.57 pWHz$^{-1/2}$, which is comparable with those of the cutting-edge solid-state photo-detectors) facilitated target-structure-

independent multi-view stereoscopic sensing. Indeed, two types of multi-functional modules specialising in curvilinear solid or hollow structure were designed with the reflective capsule photo-monitoring of columnar objects and transmissive endoscopy of cylindrical objects. These not only performed non-destructive multi-view stereoscopic inspections of actual defective industrial components (beverage bottles, water, or gas pipes), but also demonstrated remote arbitrary hierarchical image extraction of a 3D multi-layered structure via multi-frequency band scanning (sub-THz and NIR). Furthermore, the unification of free-form photo-monitoring modules and miniature photo-sources (six NIR LED-implanted portable 360°-view stereoscopic photo-imager) by 3D printing and UV laser processing, played an important role in making the system mobile for target-location-independent operations. This built-in implementation technique finally resulted in the driving of a robot-assisted highly manoeuvrable sensing system. Consequently, the robot-assisted photo-source and imager built-in multi-axis movable PTE monitor arm, which represents a consolidation of functionality, led to the non-destructive unmanned remote high-speed omni-directional photo-monitoring of a miniature aerial defective winding road-bridge. With the proposed concept, we took an initiative in modelling a roadmap for universal safety and quality assurance networks (with arbitrary target structure and location). We further expect that a variety of high-usability packaged photo-monitoring solutions, for the industrial, medical and agricultural fields, are feasible by collaborating with relevant state-of-the-art studies on roll-to-roll module printing methods[64], flexible and elastic high-density pixel integration[65] and artificial intelligent image processing[66].

## Methods

**Steady-state thermal simulation**. For the design of the flexible CNT-film broadband PTE photo-imagers, we used ANSYS to perform a steady-state thermal analysis and investigate the heat distribution and temperature gradient, across the CNT film during light absorption. We applied heat flows on the analysis models based on the assumption that CNT films are efficient at photo-absorption. By setting the values of the thermophysical property of the CNT-film channel and the stretchable electrode, the finite element method was used to solve the following equation:

$$\rho C \frac{\partial T}{\partial t} = k \left( \frac{\partial^2}{\partial x^2} + \frac{\partial^2}{\partial y^2} + \frac{\partial^2}{\partial z^2} \right) T + Q \qquad (2)$$

where, $\rho$, $C$, $k$, $T$ and $Q$ are the density, heat capacity, thermal conductivity, absolute temperature and total heat flow, respectively[47]. The thermophysical property values of the CNT film are $k$: 10 W/mK[46], emissivity $\varepsilon$: 0.9 (calculated from Kirchhoff's law of thermal radiation: $\varepsilon$ = Absorptivity $\alpha$). We set the outside temperature to a constant of 295 K.

**Selective suction CNT-film patterning**. We performed suction CNT-film filtration for selective channel patterning. A semiconducting and metallic mixed-type SWCNT solution (Zeon Co.), was dripped on a membrane filter (70 μm thick, 200-nm pore diameter, C020A025A, ADVANTEC Ltd.), which was covered in a polyimide mask (5 μm thick, Kapton, DU PONT-TORAY Co.) as shown in Supplementary Fig. 10, and then vacuumed (Vacuum pump: MVP015, Pfeiffer Vacuum Technology AG). The polyimide mask was processed by a UV laser (wavelength $\lambda$ = 355 nm, LWL-3030-T06, SIGMAKOKI Co.), and the minimum processing resolution was 10 μm. The typical channel length, channel width, channel thickness, and channel pitch were 5 mm, 1 mm, 1 μm and 1 mm, respectively.

**Chemical-carrier doping**. Liquid coating chemical doping was used as the carrier injection method for the CNT film. A mixed dopant of NaOH aqueous solution (TOKYO CHEMICAL INDUSTRY Co., LTD) and 15-crown 5-ether aqueous solution (TOKYO CHEMICAL INDUSTRY Co., LTD) was dripped onto an originally P-type CNT film. The anion (Na$^+$) was trapped by the crown ether with respect to the freely acting cation (OH$^-$), and locally changed to an N-type at the point of application[46]. Doping was performed under atmospheric pressure and room temperature, and the device can be operated as a PTE sensor immediately after drying.

**Stretchable electrode**. Silver-nanowire-based conductive elastic paste (ELE-PASTE NP1 (TR70901), TAIYO INK MFG Co., LTD), was used for the readout electrode of the flexible CNT-film PTE photo-imager. The stretchable paste was annealed for 10 min at 100 °C after using it to pattern the edge of the CNT-film channels. The electrode exhibited more than 50% omni-directional stretchability. The typical dimensions of stretchable readout electrodes were 5 mm length, 0.2 mm width and 100 μm thickness for the condition of Supplementary Fig. 6c, and 5 mm length, 1 mm width and 1 mm thickness for the condition of Supplementary Fig. 6d.

**Substrate fabrication with a 3D printer**. The hemicylindrical substrate with a window for the reflective multi-view capsule PTE imager, the column supporter for the transmissive multi-view PTE endoscope, and the compact cylindrical substrate with multiple windows for the photo-source-implanted portable 360° around-view PTE imager, was fabricated by a 3D printer (Value3D Magix MF-2200D, MUTOH INDUSTRIES LTD). Both the shapes of the substrate and the features, such as the windows, were designed on 3D CAD software (AUTODESK TINKER CAD). The minimum processing accuracy was 50 μm in the XY-axis and 100 μm in the Z-axis. Polylactic acid and acrylonitrile-butadiene-styrene were used for the printing resins. The size of each component was as follows: the hemicylindrical substrate (14-mmφ, 3-mm-thickness, 30-mm-height, 2-mm-square window), the column supporter (12 mmφ, 100 mm height), the cylindrical substrate (15-mmφ, 3-mm-thickness, 30-mm-height, six 2-mm-square windows).

**Current-voltage measurements**. To measure the DC current-voltage characteristics of the CNT-film PTE sensor, we used a current amplifier (1211, TOYO Co.) and a programmable voltage source (3245 A, Hewlett Packard Co.). The signal was recorded on a digital multimeter (34410 A, KEYSIGHT TECHNOLOGIES Inc.) and controlled by a LabVIEW program on a PC via a GPIB cable. The measurement resolution of the digital multimeter was 100 nV, and the CNT-film PTE sensor and the digital multimeter were directly connected without passing them through an amplifier, for measuring the photo-induced PTE DC-voltage responses.

**NEP evaluation**. To evaluate the photo-detection sensitivity of the CNT-film PTE sensor, we used the NEP, which is expressed by the following equation:[47]

$$\text{NEP} = \frac{V_{\text{Noise}}}{V_{\text{Sensitivity}}} = \frac{\sqrt{4k_{\text{B}}TR}}{S_{\text{Eff}} \times \triangle T} \times P_{\text{Eff}} \quad (3)$$

Here, $V_{\text{Noise}}$, $V_{\text{Sensitivity}}$, $k_{\text{B}}$, $T$, $R$, $S_{\text{Eff}}$, $\triangle T$ and $P_{\text{Eff}}$ are the noise voltage spectral density, normalised photo-induced PTE DC-voltage response, Boltzmann constant, electrical resistance, effective Seebeck coefficient ($S_{\text{P-CNT film}} - S_{\text{N-CNT film}}$), and effective power of the light irradiation on the photo-detection interface, respectively. We used a lock-in amplifier (LI-575, NF Co.) to measure the noise spectrum of CNT-film PTE sensors. The measurement range of the noise voltage was from 1 Hz to 100 kHz.

**PTE conversion sensitivity evaluation**. We used the typical dimensionless ZT values represented by the following equation to evaluate the TE conversion performance.

$$\text{ZT} = \frac{S^2 \sigma}{k} T \propto \frac{S^2 \sigma}{k} \quad (4)$$

Here, $S$, $\sigma$, $k$ and $T$ are the Seebeck coefficient, electrical conductivity, thermal conductivity and absolute temperature, respectively. We measured the change ratios of the Seebeck coefficient, photo-absorbance in the THz frequency band, electrical resistance, and thermal conductivity, in response to the concentration of the N-type chemical-carrier dopant on the CNT films. Based on the following equation, the optimal N-type chemical-carrier doping concentration for the PTE conversion performance was selected by comparing the theoretical values with the experimental photo-detection sensitivities (NEP values):

$$\text{NEP}^{-1} \Longleftrightarrow \frac{S^2}{R \times k} \times A \quad (5)$$

where $S$, $R$, $k$ and $A$ are the Seebeck coefficient, electrical resistance, thermal conductivity and photo-absorption ratio of the chemically N-type-doped CNT films, respectively. The measured values of each parameter were normalised by those of the undoped CNT films, for tuning the N-type chemical-carrier doping as shown in Fig. 1d and Supplementary Fig. 4e.

**Measurement of the Seebeck coefficient**. To measure the change ratio of the Seebeck coefficient in response to the concentration of the N-type chemical charrier dopant on the CNT films, we employed a micro-ceramic heater (MS-M1000, SAKAGUCHI E.H. VOC CO.), a K-type thermocouple (T-35K, SAKA-GUCHI E.H. VOC Co), and the aforementioned digital multimeter. One end of the channel was locally heated (+5 °C) by the micro-ceramic heater, and the corresponding voltage response was recorded on the digital multimeter, via the K-type thermocouple probed at both ends of the channel. We calculated the Seebeck coefficient of the target CNT film using the Seebeck coefficient of each Alumel

terminal, and Chromel terminal of the K-type thermocouple ($S_{\text{Alumel}} = -18$ μV/K, $S_{\text{Chromel}} = 22$ μV/K). We then measured the Seebeck coefficient measurement in a dark room under atmospheric pressure and room temperature.

**Measurement of the thermal conductivity**. We used a Xenon laser flash (LFA 467 HyperFlash, NETZSCH Japan K.K.) to measure the change ratio of the thermal conductivity, in response to the concentration of the N-type chemical-carrier dopant on the CNT films. The measured values of the chemically N-type-doped CNT films were normalised by that of the undoped CNT film. We prepared semiconducting and metallic mixed-type 25-mmφ SWCNT film with a thickness of 30 μm, for measuring the film's thermal conductivity. The thermal conductivity measurements were performed in the temperature range of −100 to 500 °C, a time acquisition speed of 2 MHz, a thermal diffusivity range of 0.01–2000 mm²/s, and a thermal conductivity range of 0.1–4000 W/mK. Here, the thermal conductivity of the undoped CNT film has been reported to be about 10 W/mK[46].

**Measurement of the absorbance spectrum**. We used terahertz time domain spectroscopy (THz-TDS, TAS7x00TS, ADVANTEST Co.) to measure the change of photo-absorption ratio change, in response to the concentration of the N-type chemical-carrier dopant on the CNT films, in the terahertz frequency band. To tune the N-type chemical-carrier doping (Fig. 1d and Supplementary Fig. 4e), photo-absorption ratio values (at $\lambda = 577$ μm/300 μm) obtained by the THz-TDS measurement, were reflected in accordance with the used THz irradiation frequencies. The THz pulse with a frequency bandwidth of 0.5–7 THz was used to emit radiation from a Cherenkov-type THz wave generator, using a nonlinear optical crystal ($\text{LiNbO}_3$) waveguide and irradiate the target sample. The transmitted pulse was guided and focused onto the photoconductive antenna (PC detector) mounted on a hyper-hemispherical silicon lens. The absorption spectrum was measured under a time resolution of 2 fs, a frequency resolution of 3.8 GHz, and a scan range of 262 ps.

**Photo-source**. In this study, we used six types of photo-sources: frequency multiplier in the sub-THz and THz bands ($\lambda = 1.15$ mm, 10-mmφ-collimated irradiation, Amplifier Multiplier Chain 573/$\lambda = 577$ μm, 4-mmφ-collimated irradiation, 520–532 GHz Custom Modular Tx-Transmitter/$\lambda = 300$ μm, 4-mmφ-collimated irradiation (also available under $\lambda = 909$ μm, 10-mmφ-collimated irradiation), Transmitter Tx261, Virginia Diodes, Inc.), a gas laser ($\lambda = 10.3$ μm, 20-mmφ-collimated irradiation/1-mmφ-fibre irradiation, $\text{CO}_2$ Laser L4, Access Laser Co.) and a radiator (broadband, 600 W, Carbon Heater DCT-J066, YAMAZEN Co.) in the FIR band, LED in the NIR band ($\lambda = 870$ nm, 5-mmφ-collimated irradiation, L12170, Hamamatsu Photonics K.K.). A conical horn antenna was attached to the frequency multiplier, and collimation lenses were built into other photo-sources.

**Multi-pixel photo-induced PTE voltage signal readout**. We used a multiplexer data logger (34980A-34923A/T, KEYSIGHT TECHNOLOGIES Inc.) to readout the multi-pixel photo-induced PTE DC-voltage signals. Up to eight terminal blocks can be mounted on one multiplexer data logger, and there can be up to 80 elements per a terminal block. The fastest readout speed is 500 channels/s; however, there is a trade-off between readout speed and readout resolution. In this study, the readout speed and resolution were, respectively, set to 50 channels/s and 100 nV, respectively. Supplementary Fig. 11b shows that the X-axis and Y-axis of the obtained PTE images correspond to the scan direction and pixel-array direction, respectively. By performing multi-pixel PTE-array scanning, the acquisition time of 2D images becomes over 20 times shorter than that conducted by utilising a single-pixel of the CNT-film PTE photo-imager, with the same spatial resolution. In this work, the number of array pixels is: 20 for the reflective multi-view capsule PTE imager, 18 for the transmissive multi-view PTE endoscope, 40 for the self-driving multi-view PTE endoscope, 24 for the photo-source-implanted portable 360°-view PTE imager.

**Stepping motor scanning**. In this work, scanning measurements were performed by connecting the CNT-film PTE photo-imaging modules with the aforementioned digital multimeter, multiplexer data logger, and target samples with a digital stepping motor (MORTARIZED STAGE, SIGMAKOKI Co.). The minimum stepping resolution is 500 nm, and the scan frequency was set to 10 Hz, when utilising the stepping motor and digital data loggers together. For the single-pixel XY imaging, the photo-induced PTE DC-voltage response at each point was recorded on the digital multimeter, while controlling the XY position of target objects at every 100 μm. For the multi-pixel-array scan imaging, the photo-induced PTE DC-voltage responses at each point were recorded on the multiplexer data logger, while controlling the one-axis position of target objects at every 100 μm. Here, the scan interval and speed can be controlled flexibly in response to the used photo-sources. For measurements with longer wavelength photo-sources (e.g. MMW ~ THz), the scan interval can range from sub-mm to a few mm, longer than the target wavelengths and the scan speed could be up to a few cm/s.

**Operation of the reflective multi-view stereoscopic capsule PTE imager**. The target columnar object connected to an XYZ-axis stepping motor was inserted inside the arc of the reflective multi-view stereoscopic capsule PTE imager. The capsule photo-imager module was fixed with an optical jig, and the spatial positions of the target columns including scanning motion, were remotely controlled by the stepping motor. Each X/Y-axis of the obtained reference image shown in Supplementary Fig. 13b corresponds to the pixel-array direction and the scan direction, respectively. While the interior wall of the 3D-printed hemicylindrical substrate was covered in a freely attachable CNT-film photo-imager array sheet, except for the window frame, the dark-coloured area ranging from 60° to 120° in the aforementioned reference image corresponded to the effective photo-monitoring region. Approximately 1/6th of the outer surface (60°/360°) of the target column was illuminated with the collimated light. The spread angle of the corresponding reflection signal reached ~60°. Each readout photo-induced PTE DC-voltage signal was calibrated by subtracting that acquired without target columns, to suppress the effect of the external collimated light irradiation on the pixels adjacent to the window frame. Each 3D image was expressed by restoring the obtained corresponding reflective 2D images, in accordance with the curvature on the semicircular outer surface of the target columns. Note that the reflection signals in the direction towards the window frame were thinned out.

**Operation of the transmissive multi-view stereoscopic PTE endoscope**. The transmissive multi-view stereoscopic PTE endoscope was inserted inside of the target cylindrical object, which was connected to an XYZ-axis stepping motor. The endoscopic photo-imager module was fixed with an optical jig, and the spatial positions of target cylinders, including scanning motion were remotely controlled by the stepping motor. Each X/Y-axis of the obtained reference image shown in Supplementary Fig. 15a corresponds to the pixel-array direction and the scan direction, respectively. While 2/3rd of the outer surface of the 3D-printed columnar supporter was covered in the freely attachable CNT-film photo-imager array sheet, the dark-coloured area ranging from 80° to 160° of the aforementioned reference image corresponds to the effective photo-monitoring region. Here, we irradiated the collimated light, meaning that ~2/9th of the outer surface of the target cylinder was illuminated with the collimated light (80°/360°). Each 3D image was expressed by restoring the obtained corresponding transmissive 2D images in accordance with the curvature of the semicircular outer surface of the target cylinder.

**Operation of the photo-source-implanted portable 360o-view stereoscopic PTE imager**. We first prepared a 15-mmφ compact cylindrical substrate. Six NIR LEDs were, respectively, implanted into the window frames of the 3D-printed substrate. CNT-film channels were printed on a membrane filter via the aforementioned selective suction CNT-film patterning method as follows: (4-pixel array → 2-mm-square window frame) × 6. Therefore, the CNT-film array sheet was rolled in a 360° cylindrical shape after the liquid coating of the N-type chemical-carrier doping, and the wiring process via the stretchable electrodes. Finally, the built-in implementation of the 360° cylindrical CNT-film photo-imager array sheet onto the interior wall of the cylindrical substrate equipped with NIR LEDs was performed. The target columnar object connected to an XYZ-axis stepping motor was inserted inside the photo-source-implanted portable 360°-view PTE imager. The spatial positions of target columns including scanning motion were remotely controlled by the stepping motor. Each X/Y-axis of the obtained reference image shown in Supplementary Fig. 17 corresponds to the pixel-array direction and the scan direction, respectively. Each readout photo-induced PTE DC-voltage signal was calibrated by subtracting that acquired without target columns, to suppress the effect of the external collimated light irradiation on the pixels adjacent to the window frames. Each 3D image was expressed by restoring the obtained corresponding reflective 2D images, in accordance with the curvature of the outer surface of the target columns. Note that the reflection signals in the direction of the window frames were thinned out.

## Data availability
The data that support the figures in this paper and other findings of this study are available from the corresponding author upon reasonable request.

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

## Acknowledgements

We thank ZEON Corporation for providing the carbon nanotube films. We also appreciate assistant professor Yoshiyuki Nonoguchi (Nara Advanced Institute of Science and Technology, Japan) for his support in measuring the thermal conductivity of carbon nanotube films. This work was supported in part by the Miral Program, the A-STEP Program, the Center of Innovation Program from the Japan Science and Technology Agency, the Toray Science Foundation, the Tateishi Science and Technology Foundation, JSPS KAKENHI (Grant Numbers JP17H02730, JP18H03766, JP19K22099, JP19H02199, JP19H04539, and JP21H01746) from the Japan Society for the Promotion of Science and DLab Challenge from Tokyo Institute of Technology.

## Author contributions

K.L. performed the device fabrication, experiments, simulations and wrote the manuscript. R.Y., R.U., M.S. and Y.T. assisted with the device fabrication and experimental setup. Especially, R.Y. helped with the transmissive multi-view stereoscopic PTE endoscopy. R.U. performed the 3D printing. R.U. and M.S. provided assistance with the self-driving transmissive multi-view stereoscopic PTE endoscopy. M.S. and Y.T. aided in measuring the PTE properties of the CNT-film photo-imager. D.S. offered advice on the experimental system and data analysis. Y.K. conceived the study, participated in its coordination, and assisted in writing the manuscript. All authors made significant contributions to the experiments, analysis of the data and writing of the manuscript.

## Competing interests

Y.K. and D.S. have filed patent applications related to this work. The other authors have no competing interests.
