## [Peer Review File · Nature Communications]

REVIEWER COMMENTS

Reviewer #1 (Remarks to the Author):

Major revisions.

This manuscript presents a robot-assisted, photo-source-imager-implanted, reflective/transmissive-modes switchable, and multi-view stereoscopic sensitive broadband photo-monitoring platform. A multifaceted photo-thermoelectric device design on flexible carbon nanotube films verifies the prototype driving of non-destructive, target-structure-independent, free-form multi-view stereoscopic examinations on actual 3D industrial components. I recommend to accept this manuscript after the authors can address the following major revisions:

(1) The CNTs were utilized in this manuscript, but which CNTs have been chosen: single wall-CNT or mutli wall CNTS? which has the better PTE? They need to compare the two materials.

(2) Some related papers about the photo-thermoelectric effect should be cited: especially, the words "Yang et al. has firstly developed the thermo-phototronic effect [1], which is based on the combination of thermoelectric effect with the photoelectric effect in some semiconductor materials" should be added in the introduction section.

[1] Thermo-Phototronic Effect Enhanced InP/ZnO Nanorod Heterojunction Solar Cells for Self-Powered Wearable Electronics, DOI: 10.1002/ADFM.201703331

(3) The figures are too many. They need to integrate the figures and also make the figures to be clear.

(4) As the sensor system, what are the response time and the recovery time for the used PTE effect? why the authors used the PTF effect not other methods? What is the advantage of this technology?

(5) The authors used the robot-assisted method in this manuscript. They need to describe more about why they used this method? Is it necessary?

Reviewer #2 (Remarks to the Author):

In this manuscript the authors developed a new design of a broadband monitoring system with multi-view stereoscopic and transmission/reflection mode switchable properties. They fabricated an all-in-one source-sensing module by a built-in photo source and a flexible single-walled carbon nanotube film with high photo-thermoelectric properties. In addition, they demonstrated their platform with the assistance of a robotic arm, showing good results for an image sensor in omnidirectional.

I think their study shows very important results for a monitoring platform in the next generation. However, there are some controversies they should address to make this manuscript more valuable.

1. I think English correction in the manuscript are should be needed. Moreover, the sentences should be expressed more concisely and clearly.

2. The authors argued that single-walled carbon nanotubes themselves already show a sufficiently high photo thermoelectric effect. Then, why is the PN junction in the single-walled carbon nanotube film required for sensing module? I think the authors should elaborate on the reason for using the single-walled carbon nanotube film with PN junctions as a photo-imager in the main text.

3. In Fig. 1b, 1d and Fig. S2c, the units should be indicated.

4. They demonstrated the photo response of their module in terms of the doping concentration, substrate materials, bending, and sample configuration. However, it is well known that the focus of light can affect the photo response and the thermal gradient. And the module is in a cylinder for the constant sensing distance. So, I think the photo response of the sensor is strongly affected depending on the radius of cylinder and the shape of a target object. Therefore, the author should demonstrate how the images obtained from their sensing module can be reliable against these factors.

In my conclusion, I recommend to publish this manuscript in Nature Communications if mandatory revisions are made by the authors.

Reviewer #3 (Remarks to the Author):

Review of Nature Communications manuscript NCOMMS-20-37017

"Robot-assisted, source-camera-coupled multi-view broadband imagers for ubiquitous sensing platform" by Kou Li , Ryoichi Yuasa , Ryogo Utaki , Meiling Sun , Yu Tokumoto , Daichi Suzuki , and Yukio Kawano

This manuscript builds on a multi-year effort that has explored the use of the photo-thermoelectric (PTE) effect in flexible films of randomly oriented single wall carbon nanotubes (CNTs). The previous development has optimized this performance (e.g. ref 14,19,31 and several others). The perspective here is partly from the THz region where many different detector technologies exist, but the much sought-after concept of a multi-pixel THz camera has still not been realized to be equivalent optical cameras. The Si/Ge detector technology is still not at that stage (ref. 37, also Pfeiffer et al, IEEE Microwave Magazine, Sept. 2019, p. 32), it uses linear arrays that are scanned in the perpendicular direction. Countless papers present THz picture of paper clips such as that S8d taken with a single THz detector of some sort and an active THz source. The CNT film used in the present manuscript is a lower cost technology but quite appropriate for the tasks it is intended for, due to its flexibility, simple doping to produce p-n-junctions with high PTE sensitivity, and ability to work from the THz to the visible region and to use built-in sources. The technology requires mechanical scanning although with moderate size linear detector arrays. The approach appears unique and has a good probability of being transformed into practical applications. This reviewer thus recommends that the manuscript be published with a few minor revisions:

- 1) Imaging speed: The last sentence, first paragraph, page 1, states that high-speed imaging is possible. However, this is never quantified. The manuscript should give examples of the actual time some of the images took to complete and future prospects for the speed. Page four, right column mentions a 24 times increase in speed by using active built-in sources; 24 times from what to what?
- 2) Page 2: An NEP of 8.57 pWHz-1/2 is mentioned for 10. 3 um wavelength. The NEP for sub-THz, THz and the NIR should also be quoted here (Fig. S2 has about 30 pWHz-1/2 at 1 THz).
- 3) Figure S8 has examples of imaging resolution paper clips etc. The resolutions appears worse than in ref. 19.
- 4) Figure S2a: The Seebeck coefficient here is about 40 uV/K. Other papers mention that it is 100 uV/K. Comment?
- 5) The sub-THz source is a commercial Virginia Diodes source that is not very compact and also quite expensive. What should be used in actual applications? Built-in sub-THz/THz sources (page 4) also appears quite optimistic given their typical sizes and costs. IMPATT/Gunn sources use quite large cavities etc. Please comment on this!

A small number of suggestions for the language:

Abstract, line 4: ...thermoelectric device design based on flexible ...

Abstract line 8: .."driving" can maybe be dropped

Introduction, first line: "technique" should be "techniques"

Outlook and conclusion: "brushing up" : "enhancing" or similar would be better

Second paragraph, line 6: "have been in constant demand" would be better English

Authors' response to the reviewer reports:

We thank the reviewers for reading our manuscript and providing their valuable comments, which were very helpful for improving the paper. We have corrected our manuscript according to the comments of the reviewers.

Responses to Reviewer 1

1- *The CNTs were utilized in this manuscript, but which CNTs have been chosen: single wall-CNT or multi wall CNTS? which has the better PTE? They need to compare the two materials.*

We employed semiconducting-metallic mixed-type single-walled CNT (SWCNT) films. We also added a simple comparison of PTE properties between the commonly used semiconducting and metallic mixed-type SWCNT film channel, the multi-walled CNT (MWCNT) film channel, and the semiconducting-separated SWCNT film channel (Supplementary Fig. 2). Based on the obtained results, the semiconducting and metallic mixed-type SWCNT film is thought to be the most suitable material for the flexible broadband photo-imager among the potential candidates.

While SWCNTs are categorised into semiconducting-separated, metallic-separated, and semiconducting-metallic-mixed, *semi-* and *mixed-*SWCNTs are potentially suitable as the PTE sensor channel, because of their higher Seebeck coefficients. Although the semiconducting-separated SWCNT film exhibits the highest Seebeck coefficient among the three (Supplementary Fig. 2a, 100 $\mu\text{V/K}$ for undoped state), its lower photo-absorbance characteristic (Supplementary Fig. 2c, 3.7 % for semi and 76 % for mixed at $\lambda = 577 \mu\text{m}$) resulted in weaker photo-response compared with that of mixed-SWCNT films (Supplementary Fig. 2d). On the other hand, MWCNT films show higher photo-absorption characteristics than mixed-SWCNT films (Supplementary Fig. 2c, 94 % at $\lambda = 577 \mu\text{m}$). However, the higher Seebeck coefficients of mixed-SWCNT films (Supplementary Fig. 2a, 46 $\mu\text{V/K}$ for SW and 18 $\mu\text{V/K}$ for MW) brought superior photo-detection performance over MWCNT films (Supplementary Fig. 2d). Based on the above discussion, we have written the following explanation in the main text.

“Here, a semiconducting and metallic mixed-type SWCNT film was employed as the photo-imager channel material because of its superior PTE properties compared to those of multi-walled or semiconducting-separated single-walled CNT films (Supplementary Fig. 2).”

2- *Some related papers about the photo-thermoelectric effect should be cited: especially, the words "Yang et al. has firstly developed the thermo-phototronic effect [1], which is based on the combination of thermoelectric effect with the photoelectric effect in some semiconductor materials" should be added in the introduction section.*

[1] *Thermo-Phototronic Effect Enhanced InP/ZnO Nanorod Heterojunction Solar Cells for Self-Powered Wearable Electronics, DOI: 10.1002/ADFM.201703331*

We sincerely appreciate your suggestion. We have written the following explanation in the Introduction section.

“In a related study, Yang et al. developed the thermo-phototronic effect²⁵, which is based on the combination of thermoelectric effect with the photoelectric effect in some semiconductor materials.”

3- *The figures are too many. They need to integrate the figures and also make the figures to be clear.*

As the referee suggested, the figures in the original manuscript need to be integrated. Therefore, we simplified and combined some of these for the revised manuscript.

We cut some schematics and 2D PTE images in the main figures (Schematic: Figs. 2a and 3b–c, 2D images: Figs. 2c, 3b–c, 4b–c, 7c, of the original manuscript). We have also simplified some schematics, photographs, and 2D PTE images in the main figures (Figs. 2b–c, 3b–c, 4a, 5a–b of the original manuscript). Furthermore, simplified figures have been integrated: “Figure 2. Reflective multi-view stereoscopic capsule PTE imager. (a combination of Figs 2–3 of the original manuscript)”, “Figure 3. Transmissive multi-view stereoscopic PTE endoscope. (a combination of Figs 4–5 of the original manuscript)”, “Figure 4. Photo-source-implanted portable 360°-view stereoscopic PTE imager for the robot-assisted monitoring applications. (a combination of Figs 6–7 of the original manuscript)”. A 3D-reconstructed PTE image has been added as Fig. 3e, showing the self-driving multi-view endoscopy of the defective miniature L-shaped tunnel.

4- *As the sensor system, what are the response time and the recovery time for the used PTE effect? why the authors used the PTF effect not other methods? What is the advantage of this technology?*

We have measured the time constant of the used PTE sensor using FIR irradiation ($\lambda = 10.3 \mu\text{m}$), as described in Supplementary Fig. 9. The obtained result indicates that our device can be operated in 5 ms for the response and 5.2 ms for the recovery.

Addressing the use of the PTE effect among other types of methods (e.g., photon-type devices, electronic-type devices), the uncooled broadband photo-detection operation poses promising advantages. As discussed by T. Ostuji (Trends in the research of modern terahertz detectors: plasmon detectors, *IEEE Trans. Terahertz Sci. Technol.*, **5**, 1110-1120, 2015.), both the electronic-type and PTE-type photo-detectors facilitate uncooled operation. Here, uncooled device operation is thought to be essential for practical applications in terms of the utilisation environment, being easily mobile and free from bulky cooling equipment. Meanwhile, the available frequency region for the electronic-type photo-detectors, such as CMOS sensors and Schottky barrier diodes, is limited to below 10 THz. Thus, the broadband photo-detection characteristics of PTE-type devices can be considered as a unique advantage over other methods, enabling spectroscopic usages, multi-layer monitoring including the presented demonstration (Figs. 2c–e), and so on. To support the above consideration, we have also written the below explanation in the main text.

“Among the various potential photo-detection mechanisms, the PTE technique is chosen because it facilitates the proposed scheme via advantageous broadband operation compared to electronic-type detectors, and the advantageous uncooled characteristic of the photo-imager compared to photon-type devices²⁶”

5- *The authors used the robot-assisted method in this manuscript. They need to describe more about why they used this method? Is it necessary?*

We believe that robotic operation significantly contributes to the future of a ubiquitous social sensing platform in terms of its manoeuvrability and easy access to facilitate out of the manned inspections. Although social infrastructure inspections have been conducted by manned operations, some parts of these include hazardous risks and non-sustainability. As an example, the disconnection testing of aerial power transmission lines is sometimes performed by trained specialist technicians, via reaching the testing place by themselves and confirming target conditions by eye. In addition to the dangers of working at heights, such an operation lacks versatility. By assuming the above consideration, the integration of CNT film flexible broadband photo-imagers with functional robotic configurations, such as multi-axis capability, wall-climbing, and underwater/aerial swimming, is thought to ease inspections of inaccessible facilities and enhance test accuracy. Therefore, we have written the below explanation in the Introduction section.

“The transition from manned to robotic inspection can make operation safer and more sustainable. Some examples of robotic operation include disconnection testing of power transmission lines with aerial modules, crack examination of sea bridges with wall-climbing units, and exploring cramped environments with self-driving systems.”

Responses to Reviewer 2

1- *I think English correction in the manuscript are should be needed. Moreover, the sentences should be expressed more concisely and clearly.*

The English of this manuscript has been carefully edited by an experienced English language editor, whose first language is English and who specialises in editing papers written by non-native English-speaking researchers. We have also attached a certification for English language editing.

2- *The authors argued that single-walled carbon nanotubes themselves already show a sufficiently high photo thermoelectric effect. Then, why is the PN junction in the single-walled carbon nanotube film required for sensing module? I think the authors should elaborate on the reason for using the single-walled carbon nanotube film with PN junctions as a photo-imager in the main text.*

As suggested by the referee, it is indispensable to mention the significance of the PN-junction in CNT films. In this study, the PN junction of the CNT film has been employed in the proposed device structure to facilitate higher intensity photo-response detection compared with that of channel-electrode interfaces. We have added Supplementary Figure 3 to clearly show this point.

The photo-thermoelectric (PTE) effect at the different materials interface enables us to obtain the DC voltage response by multiplying the effective Seebeck coefficient between the materials employed and photo-induced thermal gradient. Without the PN-structure in the channels, the photo-irradiation can also be detected on the CNT films-readout electrodes interfaces based on the PTE effect. However, from the viewpoint of the Seebeck coefficients ($S_{P-CNT\ film}: 47\ \mu\text{V/K}$, $S_{N-CNT\ film}: -42\ \mu\text{V/K}$, $S_{Electrode}: 1.5\ \mu\text{V/K}$), the effective Seebeck coefficient can be maximised on the PN junction compared to those of on the electrode interfaces. In detail, the absolute values for $S_{P-CNT\ film}$ and $S_{N-CNT\ film}$ are added together for the CNT film PN junction, while the effective Seebeck coefficients of the channel-electrode interfaces are suppressed and smaller ($|S_{Electrode} - S_{P-CNT\ film}|, |S_{N-CNT\ film} - S_{Electrode}|$). Thus, forming the PN junction of the CNT film channel is essential for the PTE response enhancement, where the effective Seebeck coefficient of the photo-detection interface can be maximised, and has been employed in the proposed device structure.

To support the above discussions, we have also added the following explanations to the main text.

“In this study, the PN junction of the CNT film has been employed in the proposed device structure to facilitate higher intensity photo-response detection compared with that of the channel-electrode interfaces (Supplementary Fig. 3). Each Seebeck coefficient was measured as $S_{P-CNT\ film}: 47\ \mu\text{V/K}$, $S_{N-CNT\ film}: -42\ \mu\text{V/K}$, $S_{Electrode}: 1.5\ \mu\text{V/K}$, and the effective Seebeck coefficient of the PN junction can be maximised ($S_{P-CNT\ film} - S_{N-CNT\ film}$), while those of each electrode junction are suppressed ($|S_{Electrode} - S_{P-CNT\ film}|, |S_{N-CNT\ film} - S_{Electrode}|$).”

3- *In Fig. 1b, 1d and Fig. S2c, the units should be indicated.*

We sincerely appreciate your comment. We have written the units in Figs. 1b, 1d (NEP).

For Supplementary Fig. 4c, the absorbance can be expressed as dimensionless values based on the equation:

$$A = -\log_{10} (I/I_0),$$

where A , I , I_0 are the absorbance, transmitted light intensity, and incident light intensity, respectively. Therefore, the vertical axis of Supplementary Fig. 4c is maintained as an arbitrary unit.

4- *They demonstrated the photo response of their module in terms of the doping concentration, substrate materials, bending, and sample configuration. However, it is well known that the focus of light can affect the photo response and the thermal gradient. And the module is in a cylinder for the constant sensing distance. So, I think the photo response of the sensor is strongly affected depending on the radius of cylinder and the shape of a target object. Therefore, the author should demonstrate how the images obtained from their sensing module can be reliable against these factors.*

Multi-view stereoscopic endoscopy of different size, shape, and structure of targets has been performed, as shown in Supplementary Fig. 16, to further specify the fundamental performance of the proposed scheme. The prepared targets include circular cylinders, with diameter range 15–40 mm, and non-circular cylinders (square, hexagon, star-shape). Each target and the used supporting substrate of the endoscope module were fabricated via 3D printing.

As presented in Supplementary Fig. 16e, degradation of the obtained PTE images was observed for larger target cylinders following photo-response attenuation in the longer optical path length. However, the concealed photo-absorbing impurity was somehow still identified non-destructively for the cylinder with a diameter of 40 mm, which is four times larger than that of the endoscope module. The demonstration also indicates that our endoscope module is available for the simple non-circular 3D-structured targets, such as the square and hexagon cylinders. Simultaneously, the result suggests that free-form devising of the endoscope module eases the non-destructive inspection of targets with complicated structures, including the star-shaped cylinder. In the case of using the column-shaped endoscope module against the star-shaped cylinder, the structural mismatch between these resulted in degradation of the PTE image due to the excessive and longer optical path length. Therefore, the effective use of 3D printing, where supporting resin of the sensing module is flexibly designed to adjust the target structures, facilitates operating-environment-independent systemisation, together with the devising of a freely attachable photo-imager.

Based on the above discussions, we have written the following explanation in the caption of Fig. 3a.

“Supplementary Figure 16 shows an evaluation of the fundamental endoscopy performances in response to the target size, shape and structure.”

Responses to Reviewer 3

1- *Imaging speed: The last sentence, first paragraph, page 1, states that high-speed imaging is possible. However, this is never quantified. The manuscript should give examples of the actual time some of the images took to complete and future prospects for the speed. Page four, right column mentions a 24 times increase in speed by using active built-in sources; 24 times from what to what?*

For the multi-view and 360°-view stereoscopic PTE images (Figs. 2b, 2d, 3b–c, 4b–c), the measurements were performed at a speed of 10 Hz. As an example, 360° images (Figs. 4b–c) were obtained in 25 s under a scan interval of 100 μm. Compared with single-pixel measurement, where both rotational and line scanning are required, the proposed system configuration works without rotational scanning. Therefore, 360° imaging measurements can be conducted 24 times faster than when using a single-pixel CNT-film PTE sensor with the same spatial resolution (600 s → 25 s), where “24” corresponds to the employed pixel number of our around-view module.

We note that the larger-area pixel configuration (e.g., 2D-matrix structure) is thought to accelerate the system operation speed. The current system operation speed of 10 Hz is bottlenecked by combined use with digital stepping motors. Therefore, the 360° around-view PTE images would be acquired in 200 Hz, which corresponds to the fundamental device response time (5 ms), by motionless large-area imaging.

We have also added the imaging speed of the robot-assisted measurements in the respective figure captions (Figs. 3d, 4f), and briefly clarified the relationship between the photo source wavelength and imaging speed in the Methods section. We have written the following explanations in the revised manuscript.

Figure captions:

“Here, the multi-view stereoscopic PTE images were scanned at a speed of 10 Hz (**b, d**).” — Fig. 2d.

“The multi-view stereoscopic PTE images were scanned at a speed of 10 Hz (**b–c**).” — Fig. 3c.

“The self-driving endoscope was operated at a speed of 10 mm/s.” — Fig. 3d.

“Here, the 360° around-view stereoscopic PTE images were scanned at a speed of 10 Hz (**b–c**).” — Fig. 4c.

“The multi-axis PTE monitor arm was operated at a speed of 15 mm/s.” — Fig. 4f.

Results:

“Currently, the proposed module can acquire 360° PTE images at a speed of 10 Hz in a simple and portable manner. In particular, rotational scanning is not required since the around-view on targets can be covered with multiple CNT-film pixels. Hence, the measurement time was 24 times shorter than that of single-pixel scanning with the same spatial resolution (600 s → 25 s with or without rotation, with a scan interval of 100 μm).”

Discussion:

“Larger-area pixel configuration (e.g., 2D-matrix structure) is thought to accelerate the system operation speed. The current system operation speed of 10 Hz is bottlenecked by combined use with digital stepping motors, so that 360°-view PTE images can be acquired at 200 Hz, which corresponds to the fundamental device response time (Supplementary Fig. 9), by motionless large-area imaging.”

Methods section:

“Here, the scan interval and speed can be controlled flexibly in response to the used photo sources. For measurements with longer wavelength photo sources (e.g., MMW ~ THz), the scan interval can range from sub-mm to a few mm, longer than the target wavelengths, and the scan speed could be up to a few cm/s.”

2- *Page 2: An NEP of 8.57 pWHz^{-1/2} is mentioned for 10.3 μm wavelength. The NEP for sub-THz, THz and the NIR should also be quoted here (Fig. S2 has about 30 pWHz^{-1/2} at 1 THz).*

We have written the corresponding values as follows in the caption of Figure 1.

“The experimentally obtained minimum NEP values of the proposed CNT-film photo-imager at broad wavelength bands (MMW – NIR) are as follows: 236 pWHz^{-1/2} ($\lambda = 1.15$ mm, $V = 3.26$ mV, $P_{Eff} = 256$ μW, Beam spot: 10-mm ϕ -collimated), 105 pWHz^{-1/2} ($\lambda = 577$ μm, $V = 328$ μV, $P_{Eff} = 114$ μW, Beam spot: 4-mm ϕ -collimated), 30 pWHz^{-1/2} ($\lambda = 300$ μm, $V = 598$ μV, $P_{Eff} = 5.97$ μW, Beam spot: 4-mm ϕ -collimated), 8.57 pWHz^{-1/2} ($\lambda = 10.3$ μm, $V = 13.8$ mV, $P_{Eff} = 39.3$ μW, Beam spot: 1 mm ϕ), and 11.1 pWHz^{-1/2} ($\lambda = 870$ nm, $V = 421$ μV, $P_{Eff} = 1.55$ μW, Beam spot: 5-mm ϕ -collimated), for the device with the electrical resistance of 555 Ω.”

3- *Figure S8 has examples of imaging resolution paper clips etc. The resolutions appears worse than in ref. 19.*

In Supplementary Fig. 8 of the original manuscript, the PTE image was obtained employing the multi-pixel CNT-film array photo-imager while the image in ref. 19 of the original manuscript was acquired by utilising the single-pixel PTE sensor. Regarding this point, we have added Supplementary Fig. 12 as the comparison of the PTE image resolution based on the pixel number. The resolution of images obtained by the one-dimensional multi-pixel array scanner can be improved through simple complementation.

When the multi-pixel array imager is utilised, the image can be acquired through one-axis scanning. In this scheme, the information between the pixel-pitch areas is missed, resulting in lower image resolution than that obtained by single-pixel XY scanning. However, combining the aforementioned one-axis scanning and the pixel-array shifting between each pixel-pitch enables us to enhance the image resolution. The total imaging time can be reduced to 1/n times between the single-pixel XY scanning and multi-pixel one-axis array scanning with pixel-pitch shifting, where “n” corresponds to the pixel number. Here, the configurations of the multi-pixel array imager are as follows: 10 pixels, 1-μm-thickness \times 4-mm-length \times 0.5-mm-width for each pixel, 0.5 mm for the distance of each pixel. The imaging area is 10 \times 10 mm² for each measurement. Each X/Y-axis was scanned per every 100 μm for the single-pixel measurement. The X-axis was scanned every 100 μm, and the array imager was shifted every 100 μm in the pixel-pitch (Y-axis) for the multi-pixel measurement.

4- *Figure S2a: The Seebeck coefficient here is about 40 $\mu\text{V/K}$. Other papers mention that it is 100 $\mu\text{V/K}$. Comment?*

In this study, we have employed semiconducting and metallic mixed-type single-walled CNT (SWCNT) films, which exhibit about 40 $\mu\text{V/K}$ of the Seebeck coefficient, as the channel material of the flexible broadband photo-imager instead of semiconducting-separated SWCNT films (whose Seebeck coefficient range about 100 $\mu\text{V/K}$). Regarding this point, we have added Supplementary Fig. 2 to compare the PTE properties of semiconducting and metallic mixed-type SWCNT films, semiconducting-separated SWCNT films, and multi-walled CNT films (MWCNT films, which could also be a channel material candidate). Based on the obtained results, the semiconducting and metallic mixed-type SWCNT films represent the best photo-detection performance, i.e., the highest PTE response, among the potential candidates.

While SWCNTs are categorised into semiconducting-separated, metallic-separated, and semiconducting-metallic-mixed, *semi-* and *mixed-*SWCNTs are potentially suitable as the PTE sensor channel because of their higher Seebeck coefficients. Although the semiconducting-separated SWCNT film exhibits the highest Seebeck coefficient among these three (Supplementary Fig. 2a, 100 $\mu\text{V/K}$ for undoped state), its lower photo-absorbance characteristic (Supplementary Fig. 2c, 3.7 % for semi and 76 % for mixed at $\lambda = 577 \mu\text{m}$) resulted in weaker photo-response compared to mixed-SWCNT films (Supplementary Fig. 2d). On the other hand, MWCNT films show a higher photo-absorption characteristic than mixed-SWCNT films (Supplementary Fig. 2c, 94 % at $\lambda = 577 \mu\text{m}$). However, higher Seebeck coefficients of mixed-SWCNT films (Supplementary Fig. 2a, 46 $\mu\text{V/K}$ for SW and 18 $\mu\text{V/K}$ for MW) brought superior photo-detection performance over MWCNT films (Supplementary Fig. 2d). Based on the above discussion, we have added the following explanation to the main text.

“Here, a semiconducting and metallic mixed-type SWCNT film was employed as the photo-imager channel material because of its superior PTE properties compared to those of multi-walled or semiconducting-separated single-walled CNT films (Supplementary Fig. 2).”

5- *The sub-THz source is a commercial Virginia Diodes source that is not very compact and also quite expensive. What should be used in actual applications? Built-in sub-THz/THz sources (page 4) also appears quite optimistic given their typical sizes and costs. IMPATT/Gunn sources use quite large cavities etc. Please comment on this!*

As the referee suggested, careful and proper choice of miniature sub-THz sources is important. Therefore, the reference regarding the Gunn diode was replaced with one for a Gunn diode chip. S. K. Sharma et al. (<https://doi.org/10.3938/jkps.67.619>) recently developed a compact Gunn diode packaging technique, and the whole module size can be estimated to be within the cm-scale, based on Figure 4 of the paper. Therefore, the potential miniature photo sources would be Gunn diode chips (sub-THz, MMW), well-established RTD diodes (THz) and QCL diodes (MIR), and NIR LEDs, which were employed in this manuscript.

6- *A small number of suggestions for the language:*

Abstract, line 4: ...thermoelectric device design based on flexible ...

Abstract line 8: .."driving" can maybe be dropped.

Introduction, first line: "technique" should be "techniques".

Outlook and conclusion: "brushing up" : "enhancing" or similar would be better.

Second paragraph, line 6: "have been in constant demand" would be better English.

We sincerely appreciate your helpful comments. These suggestions are reflected in the revised manuscript.

List of modifications

i) Text: Please see the yellow-coloured part of the manuscript for corrections of main text, Methods section, figure captions.

ii) Figures:

Figure 1

1. We wrote the units in a vertical axis of Figure 1b.
2. We wrote the units in the left-hand-side vertical units of Figure 1d.

Figure 2

1. We integrated Figures 2–3 of the original manuscript as Figure 2 in the revised manuscript.
2. We cut some schematics and 2D PTE images (Schematic: Figs. 2a 3b–c, 2D images: Figs. 2c, 3b–c, of the original manuscript).
3. We simplified some schematics, photographs, and 2D PTE images have been simplified (Figs. 2b–c, 3b–c, of the original manuscript).

Figure 3

1. We integrated Figures 4–5 of the original manuscript as Figure 3 in the revised manuscript.
2. We cut some 2D PTE images (Figs. 4b–c, of the original manuscript).
3. We simplified some schematics, photographs, and 2D PTE images have been simplified (Figs. 4a, 5a–b, of the original manuscript).
4. We added Figure 3e as a simple 3D PTE image reconstruction of the target tunnel.

Figure 4

1. We integrated Figures 6–7 of the original manuscript as Figure 4 in the revised manuscript.
2. We cut some 2D PTE images (Fig. 7c, of the original manuscript).

Supplementary Figure 2

1. We added this Figure, as a simple PTE-property comparison between the semiconducting and metallic mixed-type single-walled carbon nanotube (SWCNT) film, semiconducting-separated SWCNT film, and multi-walled CNT (MWCNT) film.

Supplementary Figure 3

1. We added this Figure, as a comparison of PTE responses obtained among the electrode-N type CNT film channel junction, PN junction of the CNT film channel, and P type CNT film channel-electrode junction.

Supplementary Figure 9

1. We added this Figure, as the operation speed of the CNT film PTE sensor.

Supplementary Figure 12

1. We added this Figure, as a comparison of the PTE image resolution based on pixel number.

Supplementary Figure 16

1. We added this Figure, as an imaging performance evaluation of the transmissive multi-view stereoscopic PTE endoscope against the size, shape and structure of target cylinders.

iii) Figure numbers: We corrected labelling numbers of each figure in accordance with the figure modifications.

iv) Reference numbers: We corrected labelling numbers of each reference in accordance with the figure modifications.

v) Data availability: We declared the data availability in the revised manuscript.

REVIEWERS' COMMENTS

Reviewer #1 (Remarks to the Author):

This manuscript reports a very interesting application by using photo-thermoelectric effect. The authors have achieved some revisions. I recommend it to be accepted after addressing the following minor revisions:

1. In the introduction section, there should be several cited papers after " which is based on the combination of thermoelectric effect with the photoelectric effect in some semiconductor materials.": DOI:10.1002/adfm.202010439; Advanced Materials Technologies, 2020, 5, 2000176; Advanced Electronic Materials, 2019, 5, 1900776; ACS Applied Materials & Interfaces, 2018, 10, 13712-13719.

2. Why the authors to choose CNTs not other nanomaterials for the good PTE? They need to clearly clarify it in the manuscript.

Reviewer #2 (Remarks to the Author):

I thank the authors for my asking. They carefully presented supplementary explanations for the queries of Editor and Reviewers and appropriately revised the manuscript. Therefore, I recommend to publish this manuscript in Nature communications.

Reviewer #3 (Remarks to the Author):

The manuscript now is much improved and the authors have complied with the suggestions of this reviewer

Authors' response to the reviewer reports:

We thank the reviewers for reading our manuscript and providing their valuable comments, which were very helpful for improving the paper. We have corrected our manuscript according to the comments of the reviewers.

Responses to Reviewer 1

1- *In the introduction section, there should be several cited papers after " which is based on the combination of thermoelectric effect with the photoelectric effect in some semiconductor materials.": DOI:10.1002/adfm.202010439; Advanced Materials Technologies, 2020, 5, 2000176; Advanced Electronic Materials, 2019, 5, 1900776; ACS Applied Materials & Interfaces, 2018, 10, 13712-13719.*

We sincerely appreciate your suggestion. We have referred to these four papers and cited them as ref. 26–29 in the revised manuscript.

2- *Why the authors to choose CNTs not other nanomaterials for the good PTE? They need to clearly clarify it in the manuscript.*

In this study, we have employed CNT films as the channel material for the flexible broadband photo-imager device owing to their functional physical properties. CNT films collectively demonstrate the characteristics of good mechanical strength, flexibility, and highly efficient broadband photo-absorption covering the millimetre-wave, terahertz-wave, and infrared frequency regions. Although several nanomaterials such as graphene, CNT, MoS₂, SnSe, NbS₃, and PEDOT:PSS have been studied for use in flexible photo-imagers, the unique broadband operation of CNT films renders these films advantageous in terms of functions. As MacLeod et al. (doi: 10.1039/C7EE01130J), Nakai et al. (doi: 10.7567/APEX.7.025103), and Hayashi et al. (doi: 10.7567/APEX.9.125103) discussed, CNT films exhibit Seebeck coefficients ranging from tens to hundreds of $\mu\text{V}/\text{K}$, which are comparable to or exceed the typical Seebeck coefficient values for PEDOT:PSS and other materials. Therefore, the utilisation of CNT films yields favourable PTE properties and facilitates the efficient conversion of photo-irradiation power to electrical signals. Further, the mechanical flexibility of the CNT films facilitates free-form adjustment and conformance to target structures. As mentioned in the 'Results' section of the main text, the PTE effect facilitates realisation of ubiquitous sensing platforms with uncooled broadband photo-sensing operations.

To clarify this discussion in the main text, we have added the following explanation:

“Among them, randomly stacked single-walled CNT (SWCNT) films are employed as the device channels in this study; this is because, when compared to the aforementioned materials, the CNT films collectively demonstrate high mechanical strength³⁷, flexibility³⁸, and advantageous excellent uncooled broadband photo-absorption³⁹. Moreover, the Seebeck coefficients of CNT films typically range from tens to hundreds of $\mu\text{V}/\text{K}$ ^{40–42}. Given these multi-functional properties, CNT films allow for efficient broadband operation (while showing good PTE properties), efficient conversion of photo-irradiation power to electrical signals, and

flexible adjustment and conformance to target structures.”

List of modifications for reviewer comments

- i) Text Please see the green-coloured parts of the article file.
- ii) Reference labelling number We corrected labelling numbers of each reference in accordance with the additional references.

List of modifications for editorial requests

- i) Email address of the corresponding author We have added an email address of the corresponding author after the author affiliations in the article file (yellow-coloured).
- ii) Abstract We removed abbreviations of “PTE” and “3D” in the abstract, and modified to “photo-thermoelectric” and “three-dimensional” (yellow-coloured).
- iii) Formatting style
 - (1) Speech marks We removed speech marks in the Result section of the article file, Figure legends, and Supplementary Information file.
 - (2) Bold and underlining font We modified bold and underlining font in Figure panels into normal font, except Figure panel headings. Section headings of the article file or Figure panel headings of the Figure legends are maintained as bold font.
 - (3) Mathematical terms We modified some terms such as $S_{CNT\ film}$ and P_{Eff} to $S_{CNT\ film}$ and P_{Eff} in the article file or Figure legends (yellow-coloured), and Supplementary Information file.
 - (4) Abbreviation We defined each abbreviation of Figure panels in the Figure legends (yellow-coloured).
 - (5) Arbitrary units We modified “a.u.” to “arb. units” in Figure panels.
 - (6) References We modified reference labelling numbers in the Supplementary Information file from s1–s4 to 1–4.
- iv) Author list We added the complete author list (yellow-coloured) at the end of the article file.
- v) Figures We attached ppt version of the figures in this manuscript.